

# Effect of salt seed particle surface area, composition and phase on secondary organic aerosol mass yields in oxidation flow reactors

Erik Ahlberg[1,2], Axel Eriksson[3], William H. Brune[4], Pontus Roldin[2], Birgitta Svenningsson[2]

[1] Centre for Environmental and Climate Research, Lund University, Sölvegatan 37, 223 52 Lund, Sweden
[2] Division of Nuclear Physics, Lund University, Box 118, 221 00 Lund, Sweden
[3] Ergonomics and Aerosol Technology, Lund University, Box 118, 221 00 Lund, Sweden
[4] Department of Meteorology, Pennsylvania State University, University Park, PA, United States

*Correspondence to*: Erik Ahlberg (erik.ahlberg@nuclear.lu.se)

**Abstract.** Atmospheric particulate water is ubiquitous, affecting particle transport and uptake of gases. Yet, research on the effect of water on secondary organic aerosol (SOA) mass yields is not consistent. In this study the SOA mass yields of an α-pinene and m-xylene mixture, at a concentration of 60 µg m$^{-3}$, was examined using an oxidation flow reactor operated at an atmospherically relevant RH of 60 % and a residence time of 160 s. Wet or dried ammonium sulphate and ammonium nitrate seed particles were used. By varying the amount of seed particle surface area, the underestimation of SOA formation induced by the short residence time in flow reactors was confirmed. Starting at a SOA mass concentration of ~5 µg m$^{-3}$, the maximum yield increased a factor ~2 with dry seed particles, and on average a factor 3.2 with wet seed particles. Hence, wet particles increased the SOA mass yield by ~60 % compared to the dry experiment. Maximum yield in the reactor was achieved using a surface area concentration of ~1600 µm$^2$ cm$^{-3}$. This corresponded to a condensational lifetime of 20 s for low volatile organics. The O:C ratio of SOA on wet ammonium sulphate was significantly higher than when using ammonium nitrate or dry ammonium sulphate seed particles, probably due to differences in heterogeneous chemistry.

## 1. Introduction

The atmospheric aerosol is a dynamic mixture of organic and inorganic species. A large fraction of the organic aerosol is formed as a result of atmospheric processing of volatile organic compounds (VOCs), with products condensing onto pre-existing particles, forming secondary organic aerosol (SOA) (Hallquist et al., 2009). The partitioning of semivolatile organic species from gas to particles can be either adsorptive or absorptive depending on the chemical composition and phase of the particles (Pankow, 1994). Despite the complexity of the atmospheric aerosol, SOA mass yields (mass of formed particles divided by the mass of VOCs reacted) have traditionally been parameterized in models using simplified and relatively dry laboratory experiments.

Although not always measured, water is ubiquitous in the atmospheric aerosol, influencing particle size, scattering, transportation/deposition and uptake of gases, ultimately affecting both climate and health effects of particles (Pilinis et al., 1989; Nguyen et al., 2016). The aerosol liquid water content at subsaturation of water vapour depends on the relative humidity





(RH), dry particle chemical composition and size. The most abundant inorganic aerosol compounds in submicron aerosol particles are the salts ammonium sulphate (AS) and ammonium nitrate (AN), which are mostly of anthropogenic origin (e.g. Poschl, 2005; Zhang et al., 2007). A portion of the water in particles can be regarded as anthropogenic water since both nitrate and sulphate generally increase particle hygroscopicity (Carlton and Turpin, 2013; Hodas et al., 2014), which in turn facilitates

SOA formation of water soluble organic compounds. This particle formation pathway is believed to be especially important for isoprene SOA, since isoprene's first generation oxidation products are relatively small molecules with high saturation vapour pressures compared to the oxidation products of other common SOA precursors (Carlton et al., 2009; Ervens et al., 2011; Sareen et al., 2017). Also, the electrolyte solution of ammonium sulphate and ammonium nitrate is acidic since ammonia is a weak base, which increases the reactive uptake of several SOA species (Jang et al., 2002; Gao et al., 2004).

SOA research has been substantial during the last two decades, and the effects of relative humidity and aerosol liquid water on particle yields have been investigated in numerous studies. From partitioning theory it can be shown (Seinfeld et al., 2001; Hallquist et al., 2009; Pankow, 2010) that there should be a clear SOA mass yield dependence with RH, especially at low precursor concentrations, if an activity coefficient of 1 is assumed. Hennigan *et al.* (2008) showed that in an urban region dominated by biogenic emissions, partitioning to liquid water may be a significant contributor to SOA mass. However, adding

water to laboratory oxidation experiments complicates the interpretation, since both gas (Jonsson et al., 2006; Warren et al., 2009) and particle phase (Ervens et al., 2011) chemistry may change. Furthermore, the yield variation with different RH can be NOx dependent (Ervens et al., 2011). For isoprene, both particulate water and acidity is believed to have strong effects on the SOA yield (Surratt et al., 2007; Carlton et al., 2009). Wong et al. (2015) showed that wet ammonium sulphate seed particles resulted in 60 % more isoprene SOA being formed compared to a system with dry seed particles at the same RH. However,

laboratory studies using other SOA precursors are somewhat inconsistent. Prisle *et al.* (2010) saw no influence of RH (up to near 100 %) on α-pinene ozonolysis SOA yields with ammonium sulphate seeds. Cocker III *et al.* (2001a), investigating the same system, but using dry or wet seed particles found that mass yields varied little with RH but decreased if the seed particles were wet. In similar studies the yields of m-xylene, 1,3,5-trimethylbenzene  (Cocker et al., 2001b) and toluene (Edney et al., 2000) have been found to be unaffected by the aerosol liquid water content. Lu *et al.* (2009) found no effect on m-xylene SOA

yields with wet or dry neutral seed particles, while the yield was increased with dry acidic seed particles. In contrast, other studies have found that higher RH significantly increases the SOA mass yield of toluene and xylenes (Kamens et al., 2011; Zhou et al., 2011). Also, more recently, Stirnweis *et al.* (2017) assessed the influence of NOx and RH on α-pinene SOA with different seed particles and concluded that particulate water significantly increases the organic mass yields. Further, Faust et al. (2017) found 13 and 19 % increases in SOA yield from α-pinene and toluene respectively, when SOA was formed on wet

salt particles. It can be difficult to compare the results of different SOA oxidation experiments, since the conditions used are rarely the same. SOA yields can depend on e.g. temperature, NOx concentrations, precursor concentration, oxidant exposure and type, seed particle concentration and composition.



In the following work we report SOA yields from a mixture of α-pinene and m-xylene oxidized in a PAM oxidation flow reactor (OFR) (Kang et al., 2007; Lambe et al., 2011a), in the presence of wet or dry ammonium sulphate and ammonium nitrate seed particles. In contrast to many other studies looking into the effect of particulate water on SOA mass yields, the RH was held constant at 60 %, while the seed particles were either dried below an RH of 10 % or kept in their droplet state.

Previous studies have shown that SOA produced in the reactor is similar to that produced in traditionally used smog chambers (Bruns et al., 2015; Lambe et al., 2015). The reactor can produce a more oxidized aerosol, which is strongly linked to the hygroscopicity of organic aerosols (Pang et al., 2006; Chang et al., 2010; Lambe et al., 2011b). Due to the fast processing in flow reactors, several studies have discussed the potential problem with low condensation sinks resulting in lower yields (Lambe et al., 2015; Palm et al., 2016; Ahlberg et al., 2017; Simonen et al., 2017). This effect was systematically investigated

during the course of the experiments by using different seed particle concentrations.

## 2. Methods

### 2.1 Experimental set-up

All experiments consisted of introducing a constant flow of SOA precursors with a varying concentration of seed particles into an oxidation flow reactor. The experimental set-up is shown in Fig. 1. Seed particles were formed from atomization of a ~1

g/l solution of ammonium sulphate (Sigma Aldrich, ≥99 %) or ammonium nitrate (Sigma Aldrich, ≥99.5 %) in molecular grade water. The size distribution of seed particles had a maximum volume concentration at a mobility diameter between ~150-200 nm. Although the molality of the solutions and pressure in the atomizer were similar in all experiments, the output number size distributions were not identical. The number of particles per volume unit, as measured by the SMPS increased in the following order: dry AS>wet AS>dry AN>wet AN. The RH in the reactor was chosen between the deliquescence and

efflorescence points of ammonium sulphate (Seinfeld and Pandis, 2012) and sodium chloride (experiments not reported here due to instrument failure) so that the hysteresis effect could be used to alternate between aqueous and dry particles. Ammonium nitrate have a reported deliquescence RH of 61.8 % (Tang and Munkelwitz, 1993) but efflorescence is not observed (Svenningsson, 1997; Lightstone et al., 2000). Seed particle mass concentrations of ~0-100 µg m$^{-3}$ were achieved by pulling a varying flow (0-0.7 lpm) from the atomizer through the reactor.

VOCs were introduced into the reactor using a diffusion system with thin capillaries, described in Ahlberg *et al.* (2017). VOCs were chosen to get a mix of biogenic (α-pinene) and anthropogenic (m-xylene) SOA. The flow of VOCs was held constant throughout an experiment. A relatively low, and atmospherically relevant, SOA mass concentration of ~5 µg m$^{-3}$ without seed particles was aimed for so that the nucleated particles would not be the dominant condensation sink. The VOC concentration was determined after the experiments by the liquid weight loss during four weeks. During these weeks, the evaporation rate

declined, probably due to VOC oxidation or VOC condensation inside the capillaries. Therefore the values of the first weighing was used. If the decline during the first week, prior to the first weighing, was the same as consecutive weeks, it would result



in an overestimation of 13 % in the summed yield. Since the timescales of an experiment (~8 h) was much shorter, oxidation or condensation inside capillaries is not expected to have taken place. The total concentration of VOCs at the reactor inlet was calculated to be 60 µg m$^{-3}$ (5.2 ppb α-pinene and 6.7 ppb m-xylene), hence the SOA mass yield with no seeds was ~8 %, in agreement with previous measurements of the same mixture (Ahlberg et al., 2017), albeit at slightly different VOC ratios.

## 2.2 SOA formation

SOA was produced using a PAM oxidation flow reactor, which has been extensively used in laboratory and field measurements (https://sites.google.com/site/pamwiki/). The reactor, which is a 13.2 l horizontal aluminium cylinder with passivated walls, produces very high concentrations of ozone and hydroxyl radicals (OH) from UV lights mounted inside (Kang et al., 2007; Lambe et al., 2011a). In recent years measurements and modelling have significantly advanced the knowledge of the reactor and best practices during use have been developed (Ortega et al., 2013; Li et al., 2015; Peng et al., 2015; Palm et al., 2016; Peng et al., 2016). Briefly, the reactor should not be used with too high OH reactivity input, since OH may be suppressed. The same problem may arise if the OH exposure is low due to low lamp voltage or low absolute humidity. In this work the flow was set to 5 lpm and only one lamp was used. Lamp voltage was adjusted to reach an O$_3$ concentration of 2.7-3 ppm. RH in the reactor was held constant at 60 % by PID regulation of a humidified flow. With these settings the OH exposure, calibrated off-line using 10 ppb of SO$_2$ (for detailed procedure, see Lambe et al., 2011a), was 7 x 10$^{11}$ molecules cm$^{-3}$ s, with an experimental uncertainty (1σ) of 5 %. The total OH reactivity (defined as the concentration of reactant multiplied by the OH reaction rate) was 9.4 s$^{-1}$ which is not believed to have induced significant OH suppression (Peng et al., 2015; Peng et al., 2016). The temperature increase inside the reactor due to the lamp was measured to 1-2°C prior to the experiments using a thermocouple inserted into the reactor. With an RH of 60 % at 22°C (room temperature), RH inside the reactor is expected to be 53 %, which is above the efflorescence point of ammonium sulphate at the same temperature.

Particle losses depend on reactor settings and particle sizes, but are generally low; from a few up to 10 % on a mass basis (Martinsson et al., 2015; Karjalainen et al., 2016; Ortega et al., 2016; Palm et al., 2016). Ortega *et al.* (2013) found that most of the particle losses takes place at the inlet of the reactor. In this work, losses at the inlet are not of importance, since we look at SOA formed inside the reactor only. Palm *et al.* (2016) constructed a model for the fate of low volatile organic compounds (LVOCs) in the reactor, in which four loss terms are competing: condensation onto particles, wall loss, fragmentation (assumed after reacting with OH five times) and outflowing from the reactor. The model was compared with the SOA mass yields at different seed concentrations. For the reactor settings used, the modelled LVOC fate as a function of seed particle area is shown in Fig. 2.

## 2.3 Particle measurements

After oxidative ageing in the reactor, the aerosol was dried below 30 % RH before size distribution and mass based chemical composition was measured using a scanning mobility particle sizer (SMPS, Wiedensohler et al., 2012) and an Aerodyne high



resolution time-of-flight aerosol mass spectrometer (AMS, DeCarlo et al., 2006), respectively. The SMPS consisted of a custom-built DMA and a TSI CPC (model 3010). Silica gel driers decreased the RH of the sheath flow to below 10 %. The DMA voltages were calibrated prior to experiments and the number size concentration was checked using PSL spheres. The AMS was calibrated using size selected ammonium nitrate and ammonium sulphate particles.

## 2.4 Experimental procedure and data analysis

An overview of the experiments can be seen in Table 1. Before experiments, the reactor was run without seeds or VOCs until the volume concentration was below 0.2 $\mu m^3$ $cm^{-3}$, as measured by the SMPS. Before adding VOCs, 2-5 concentration levels of pure seed particles were measured to be able to parameterize organic impurities from the atomizer as a function of salt concentration. Despite using ultrapure water and zero air (Linde, GT30000), up to 6 % of the total mass of the seed particles were organic impurities and scaled roughly in a linear way with salt ion concentration. Pieber *et al.* (2016) found interferences in the *m/z* 44 signal from reactions in the ionization region facilitated by inorganic salt particles. However, in our experiments the m/z 44 signal was only ~15 % of the total organic impurity signal. Before calculating SOA yields, the impurities were removed from the organic signal using the linear relationship with the salt ions. After adding a constant concentration of VOCs, SOA was measured at 5-8 different seed particle concentrations. For each seed type, experiments without seed particles were performed to get a base level yield. This level was relatively stable between experiments, at 5.0 ±0.5 µg $m^{-3}$ (1σ). Using data from a similar mixture in Ahlberg *et al.* (2017) the difference in base level corresponds to a difference in VOC concentrations of ±2 µg $m^{-3}$ (±3.3 %). During the dry ammonium nitrate experiment, the base level drifted from 5 µg $m^{-3}$ at the start to 6 µg $m^{-3}$ at the end of the day. For this experiment, a time adjusted base level was implemented. The adjustment translated to an increase in the SOA mass yield by at most 15 % for the lowest seed concentration, to 1 % at the highest seed concentration.

The SMPS was used to determine the particle number size distribution and total particle volume and area concentrations. To calculate the input dry seed particle surface area concentration, a parameterization from pure salt measurements was made as a function of either sulphate or nitrate concentration as measured by the AMS. The size distribution was also used to calculate the condensation sink (*CS*) (Pirjola et al., 1999). However, we use area concentration when presenting our data since this is a measurement more often used and in these experiments scaled linearly with *CS*.

AMS data was evaluated using standard AMS analysis programs (Squirrel v1.57 and Pika v1.16). Standard changes to the fragmentation table and high-resolution spectra were made, including corrections for zero air $CO_2$ concentrations and removal of organic peaks overlapping with either air or salt peaks (*m/z* 14, 16, 32, 48, 64 for AS and 14, 16, 30, 46 for AN). The ammonium nitrate calibration of the AMS was used to calculate the relative ionization efficiency (RIE) of ammonium, which was subsequently used to calculate the RIE of sulphate. The RIE of ammonium was 4, which is the default value of the AMS, but the RIE of sulphate, at 1.96, was significantly higher than the default value of 1.2. Although this means that the measured sulphate mass was decreased during analysis, it doesn't affect the seed area calculations since the parameterization and seed mass changes cancel each other. For organics, the default RIE of 1.4 was used. To evaluate the AMS collection efficiency



(CE) of the different experiments, the volume concentration as measured by the SMPS was multiplied by particle density calculated from the AMS chemical composition. A density of 1.4 g cm$^{-3}$ was used for SOA from previous parameterizations of a similar mixture (Ahlberg et al., 2017). The collection efficiencies used as a function of SOA mass fraction can be seen in Fig. S1. CE for pure salts are listed in Table 1. CE of both wet and dry ammonium sulphate increased with increasing SOA

mass fraction, likely due to decreased bounce. For ammonium nitrate, CE was roughly constant around 1. Pure SOA had a CE of 0.63±0.03 (1σ). The relatively low CE of SOA may not only be a bounce effect, since these particles were significantly smaller, with a number mode around 20 nm in mobility diameter, which to a higher degree are lost in the aerodynamic lens inlet of the AMS. The "improved ambient" parameterization was used to calculate elemental ratios (Canagaratna et al., 2015). However, the organic portion of the particles consisted of both SOA and salt impurities. To calculate the O:C and H:C ratios

of SOA only, the elemental ratios of the impurities only and their fraction of the total organics were used. This correction increased SOA O:C by ~8 % and decreased H:C by ~0.5 % for AS while for AN the change in O:C was below 1 % and H:C decreased 1-2 %.

The SOA mass yield is defined as the amount of SOA formed divided by the amount of VOCs reacted. In the oxidizing environment of the reactor, 100 % of the input VOCs are expected to react. However, comparing yields only would give a

skewed result, since small differences in base SOA level between the experiments (Table 1) give large differences in yield. Instead we compared the ratio of base level SOA mass to SOA mass at different seed particle concentrations, which is equal to the relative increase in yield (unitless). This cancels out the VOC concentrations from the calculations. The uncertainty in the yield increase was calculated from the standard deviation of the measurements. The fractional uncertainty was between 6-10 %. However, this may give a too small range. Although all flow, pressure, and OFR settings were checked, a variation

larger than the experiment standard deviation is expected since the setup was highly sensitive to small perturbations. No experiment was repeated fully, but single point replicates gave a fractional error of 14 % compared to expected values. Therefore, a conservative expected repeatability of the experiments is within 20 %.

## 3. Results and discussion

Figure 3 shows the increase in yield as a function of dry salt seed particle surface area concentration. In all experiments the

yield increased significantly with seed particle surface area, confirming previous findings (Lambe et al., 2015; Palm et al., 2016; Ahlberg et al., 2017). The error in the yields of these experiments can be calculated by normalizing the yield increase with the maximum yield increase. Doing this, all experiments follow the same trend with seed surface area, seen in Fig. 4. Also seen in Fig. 4 is the modeled bias (fraction condensed on particles), following a similar trend but slightly lower since the experimental data only considers the condensation sink of the seed particles, while in the model the produced organic sink is

also taken into account. The yield error decreases up to a condensation sink of ~0.05 s$^{-1}$ corresponding to a seed surface area of ~1600 µm$^2$ cm$^{-3}$. Above this value, the LVOC fate model (Fig. 2) also indicates a slower increase. However, while the LVOC model yield continues to increase with increased seed area concentration, the increase in the experimental yields levels



off. This could be due to increased particle losses inside the reactor, which have not been considered. The condensation sink at which the data levels off, corresponds to a lifetime ($\tau_{CS}$) of 20 s, which is similar to the residence time of the reactor short-circuit (Lambe et al., 2011a; Ahlberg et al., 2017).

The results suggest that previous measurements using similar reactors have underestimated the yield at low condensation sinks. Because the error is larger at low yields, the yield curves will have a steeper increase and reach a constant yield at lower mass concentrations. Applying corrections to previous reactor experiments relying on nucleated particles as the only condensation sink is not trivial, since the condensation sink varies with time in the reactor. However, at a similar SOA condensation sink as that used in this study (0.022 s$^{-1}$), the yields should increase by a factor 2-3 compared to when no seed is used (Fig. 3). Given the shape of the yield bias in Fig. 4, at lower concentrations (and condensation sinks) the increase should be even higher. According to the LVOC fate model a 3-fold increase in yield (yield/max yield of 0.33) corresponds to a condensation sink of ~0.006 s$^{-1}$, suggesting the effective *CS* at this mass concentration is ~1/3 (0.006/0.022) of the reactor outlet *CS* in nucleation experiments. At half of that condensation sink (0.003 s$^{-1}$), the model predicts a 5-fold increase in yield, and at 1/10 (0.0006 s$^{-1}$) the increase could be as high as a factor 45. In Fig. 5 we used the LVOC fate model to recalculate the yields of Ahlberg *et al.* (2017). The inverse of the fraction condensed on particles at 1/3 of the experimental *CS* was multiplied with the measured yields and mass concentrations. Because both x and y values increase (both SOA mass and yields change with the same factor), the change from the measurements is not as dramatic as when only looking at the absolute yield increase or if yields were plotted against reacted VOCs. At 10 µg m$^{-3}$ the increase in yield is estimated from the linear regression between the datapoints in Fig. 5 at 67 %, 80 %, 24 % and 94 % for α-pinene, m-xylene, myrcene and isoprene respectively, with the differences arising from differences in the size distribution of each SOA precursor. It is likely that our assumption that that the effective *CS* is 1/3 of the output underestimates the yield at low mass concentrations and overestimates the yield at high mass concentrations, since a higher VOC input produces a sink faster than a low input. Although the calculations may be an oversimplification, it is clear that the SOA mass yields at low mass concentrations are biased low and that seed particles have a big impact in OFR experiments.

The second main result, also seen in Fig. 3, is that the increase in yield with increased seed concentration is lower for the dry ammonium sulphate experiment. Since ammonium nitrate does not effloresce (Svenningsson, 1997; Lightstone et al., 2000) it is likely that both wet and dry AN adjusted to the RH of the reactor and thus these experiments are essentially the same. Also, while the yield increase is highest for wet AS, this experiment had a lower base level SOA mass concentration, making it harder to rank the three wet experiments. The grey area in the figure represents ±20 % of the three experiments where the seed particles did not effloresce and is added to emphasize the similarities between them. Dry ammonium sulphate was the only crystalline particle, with a yield bias of a factor ~2, while the other three experiments were similar given the experimental uncertainty, with a yield bias factor of 2.9-3.5. Hence, wet seed particles increased the yield by 45-75 %, with an average of 60 %, compared to the dry seed experiment.



The difference between wet and dry experiments can be due to either differences in partitioning, reactive uptake or both. Julin *et al.* (2014) showed that the mass accommodation coefficient of several different organic molecules is unity, regardless of the particle phase state. In their study, the condensed and gaseous phases consisted of the same molecules, which is not the case in the present study. However, as soon as a layer of organics has condensed on the crystalline AS particles, the mass

accommodation for uptake at the surface should approach unity. If an aqueous phase is to increase the yield by equilibrium partitioning, the organic molecules need to be water soluble and SOA mass concentration needs to be low enough to retain an appreciable amount in the gas phase (Hallquist et al., 2009; Pankow, 2010). Several studies have shown that organic aerosol particles may undergo liquid-liquid phase separation (Song et al., 2012b; You et al., 2012; Zuend and Seinfeld, 2012). However, the water solubility of organic molecules increases with decreasing molecular weight and increasing polarity (O:C)

(Varutbangkul et al., 2006; Massoli et al., 2010; Duplissy et al., 2011), both of which are favoured in OFR experiments compare to smog chambers. It has been shown (Song et al., 2012a, b) that liquid-liquid phase separation rarely occurs at O:C ratios higher than 0.7 in systems containing atmospherically relevant organics, water and AS. In the present study O:C was always higher than 0.7, which is seen in Fig. 6 that shows the elemental ratios in Van Krevelen space as measured by the AMS.

The elemental ratios of all experiments fall within a relatively narrow range (Fig. 6). The difference between wet and dry AN
and dry AS is similar to the difference between pure SOA on different days (white symbols), with O:C within $0.83 \pm 0.08$ (1σ) and H:C within $1.33 \pm 0.05$ (1σ). The wet AS experiment however reaches higher O:C values and spans a larger range. The O:C value increases with increasing SOA mass concentration (and seed particle concentration since these are connected), which is the opposite to what is expected since more oxidized molecules tend to be less volatile (Shilling et al., 2009). The increase is mostly due to the mass fragments with *m/z* 28 and 44. Several acid catalyzed oligomerization reactions change the
elemental ratios of SOA, but with lower O:C as a result (Jang et al., 2002; Chen et al., 2011). Also, saturated AN and AS solutions have similar pH (4.5 and 4.2 respectively according to the E-AIM model (Clegg et al., 1998)), hence there should be no big difference between the wet experiments. A more likely explanation to the change in O:C with SOA mass is the dynamics of the reactor. In these experiments, larger SOA concentrations are due to increased seed particle concentration. Increasing the seed particle number concentration also decreases the SOA mass fraction. It follows that SOA mass then is spread out on more
particles, leaving less organics per particle, which could enhance the partitioning of more volatile material to the gas phase. This would leave more time for gas phase oxidation, and consequently a higher O:C ratio, provided the molecules partition to the particles. The higher O:C of wet AS could be explained if there are differences in heterogeneous chemistry. Kroll *et al.* (2015) showed that heterogeneous OH oxidation of SOA may increase the carbon oxidation state, on the expense of SOA mass. The water uptake (growth factor) of AN and AS are slightly different at 60 % RH, with an area increase of ~1.4 and
~1.7 respectively. However the O:C increase with increasing area is much larger for wet AS than for any other experiment, hence a chemical explanation is needed. To the best of our knowledge, there are no measurements on differences between OH radical uptake on different salt solution surfaces. Wang *et al.* (2016) found that the salting out effect (pushing dissolved molecules out of the water phase) of several different organic molecules is stronger in sulphate compared to nitrate solutions.



However, the compounds used were not similar to the SOA used in this study and this effect should cause opposite results since the yield increase is slightly higher for wet AS than for the AN experiments. Takami *et al.* (1998) showed that below a pH of 7 the uptake coefficient of OH increases with acidity. Given the small difference in acidity between saturated AN and AS it is uncertain if this can explain the measurements. Wick and Dang (2006) found that solvation of OH was correlated with increasing NaCl salt concentration. A possible pathway for differences in reactive uptake of OH between the salt solutions is the reaction with $HSO_4^-$, forming sulphate radicals (Jiang et al., 1992). Sulphate radicals have been shown to be an important source of organosulphates (Noziere et al., 2010; Schindelka et al., 2013). However, sulphate from organics is indistinguishable from inorganic sulphate in the AMS mass spectra since the fragmentation patterns are the same (Farmer et al., 2010), and thus should not affect the calculated O:C ratios. If sulphate aerosols affect the organic portion, this should be seen in ambient samples. Indeed, several studies have shown that the more oxidized SOA (LV-OOA) is correlated with sulphate (Ng et al., 2011b; Zhang et al., 2011; Crippa et al., 2013; Hao et al., 2014), but a more straightforward explanation to this is the fact that both are secondary aerosol constituents.

## 4. Conclusions

Experiments were conducted with two aims: (i) to investigate the influence of an aerosol liquid water phase on SOA yields and (ii) verifying and quantifying the underestimation of SOA production in oxidation flow reactors due to limited time for condensation. It was found that in all cases there was a strong increase in yield with increased seed surface area concentration, and that the yield with wet seed particles was 45-74 % higher compared to dry seed particles. The yield increase leveled off at a dry seed particle condensation sink of ~0.05 $s^{-1}$ corresponding to a surface area concentration of ~1600 $\mu m^2 cm^{-3}$. This implies that it is crucial that the condensation sink is accounted for in OFR experiments where the absolute SOA mass is of interest.

If seed particles are used to drive the partitioning to the particle phase, the choice of seed may affect the results due to differences in heterogeneous chemistry and water uptake. This makes translation of lab results to atmospheric relevance more difficult since a much higher seed particle concentration is needed in a reactor than in the atmosphere to make the condensed fractions comparable. To further study and parameterize the effects of the condensation sink on OFR SOA, future experiments should focus on different VOC concentrations with varying seed particle surface area, as well as using different seeds or seed mixtures.

Using dry ammonium sulphate seed particles the maximum yield increase was approximately a factor 2, while all wet experiments were similar and induced an increase above a factor 3. Hence, the wet particles produced around 60 % more SOA mass. The O:C ratio increased with decreasing SOA mass fraction. Also, O:C was higher with wet AS compared to other seeds, something which needs further research to fully explain, but is likely due to heterogeneous chemistry. These results point to the importance of anthropogenic water as an important source of SOA.



*Data availability.* Data are available upon request from the corresponding author.

*Author Contributions.* EA designed the study together with BS, performed the experiments, analyzed the data and prepared the manuscript with contributions from all coauthors. AE was responsible for AMS data quality assurance.

*Competing interests.* The authors declare that they have no conflict of interest.

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

25

30



**Table 1. For each of the four experiments the table shows dry seed surface area concentration, collection efficiency (CE) of the AMS for salts without SOA, initial SOA without seed particles (two replicates where available) and maximum SOA concentrations (with seed particles).**

|  | Seed area range ($\mu m^2\ cm^{-3}$) | CE ($1\sigma$) | SOA initial ($\mu g\ m^{-3}$) | Max SOA ($\mu g\ m^{-3}$) |
|---|---|---|---|---|
| AS dry | 570-2210 | 0.57 (0.05) | 5.2, NA | 10.6 |
| AS wet | 500-2820 | 0.76 (0.01) | 4.5, 4.5 | 16.8 |
| AN dry | 40-2900 | 0.93 (0.04) | 5.0, 5.9 | 22.0 |
| AN wet | 210-3090 | 1.02 (0.07) | 5.0, 4.6 | 15.3 |





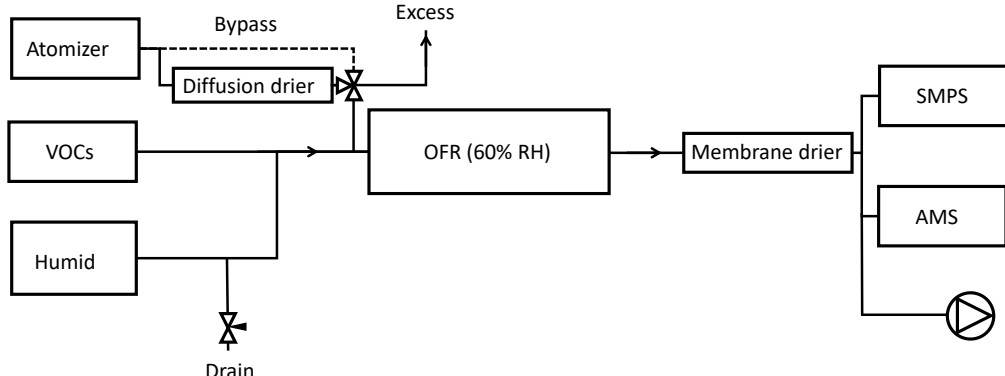

**Figure 1. Experimental set-up. Seed particles were either dried or maintained in a liquid droplet. By changing the drain flow, the flow from the atomizer was varied without perturbing the VOC concentrations in the reactor. Total flow through the reactor was 5 lpm and RH was kept constant at 60 % by varying the RH of the humid flow.**

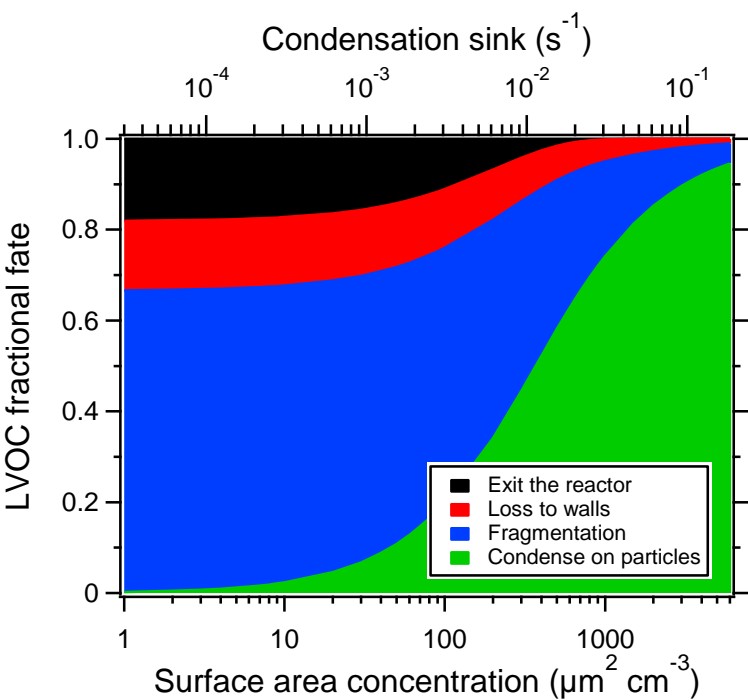

**Figure 2. The fractional fate of LVOCs as a function of particle surface area concentration, using the model of Palm *et al.* (2016). A molecule is assumed to fragment after reacting with OH five times. OH reaction rate was $1 \times 10^{-11}$ cm$^3$ molecs$^{-1}$ s$^{-1}$. For reactor settings, see text.**



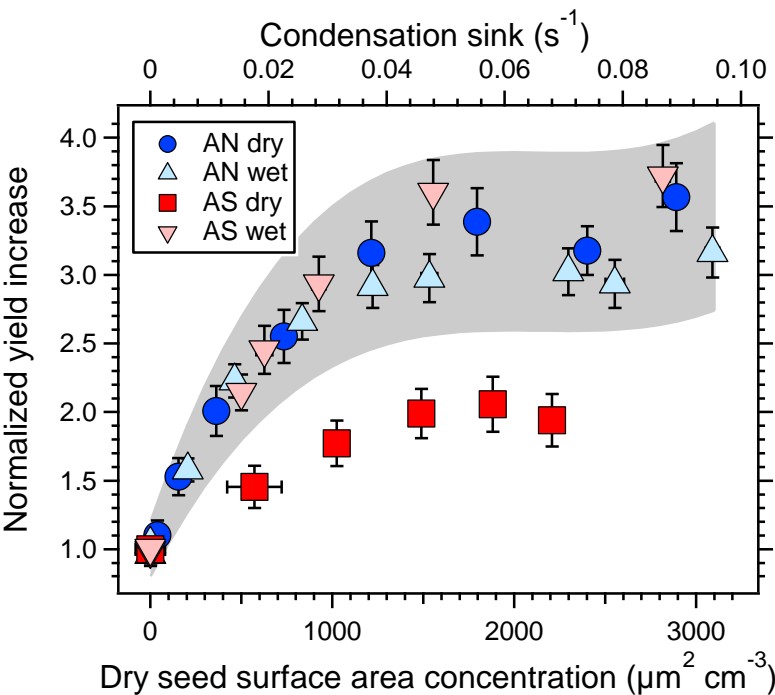

**Figure 3. The increase in yield at different dry salt seed surface areas, normalized to the yield from experiments with no seed particles. The corresponding condensation sink is shown on the top axis. Error bars denote 1σ of the measurements. The grey area represents ±20 % of the three experiments where the seeds are not effloresced to illustrate the expected repeatability of the experiments and the fact that the dry ammonium sulphate results are the only ones falling outside of this range.**

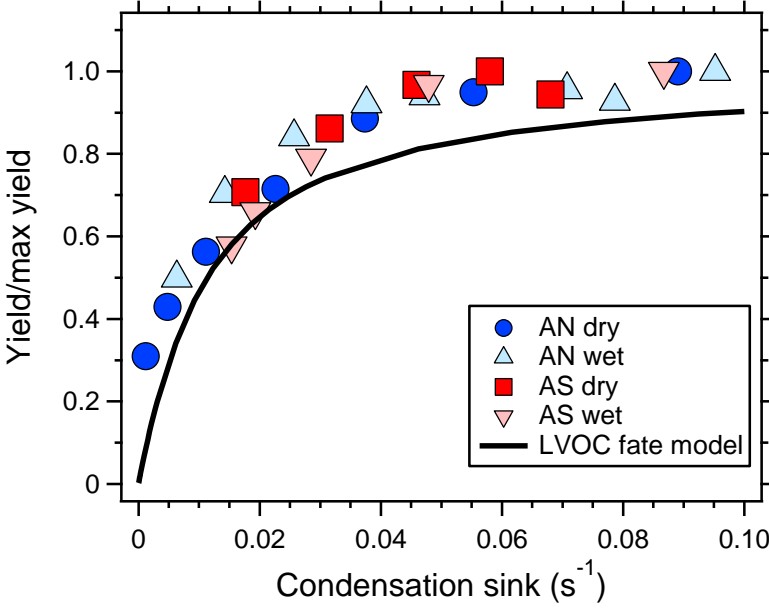




**Figure 4. The data from Fig. 3 normalized by the maximum yield increase and plotted against the dried seed particle condensation sink. The black line shows the fraction of condensed LVOCs according the model of Palm *et al.* (2016).**

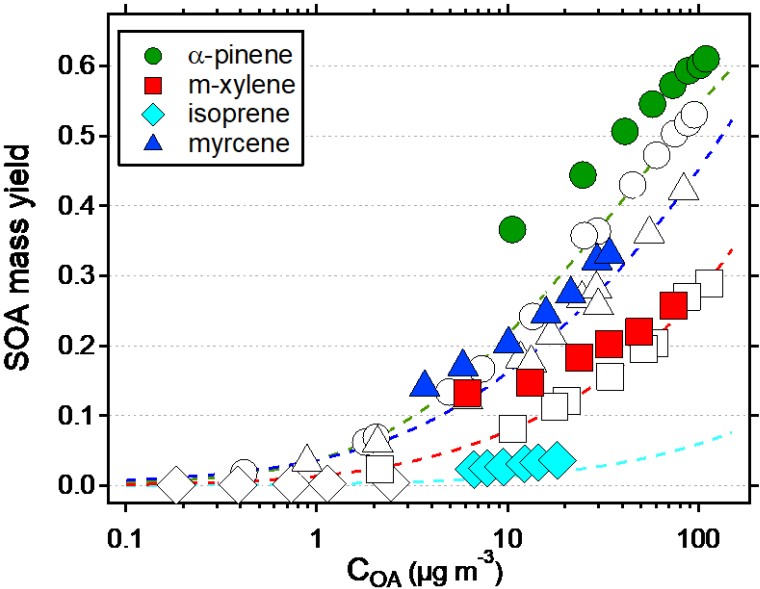

5  **Figure 5. Recalculated SOA mass yields from Ahlberg *et al.* (2017) as a function of organic aerosol mass concentration ($C_{OA}$). Assuming an effective condensation sink of 1/3 of the reactor output, the inverse of the fraction condensed in the LVOC fate model from Palm *et al.* (2015) was multiplied with the measured yields. Coloured symbols are recalculated values, and original values are represented with the same symbol but no colour. Dashed lines show the VBS models constructed from the original data.**

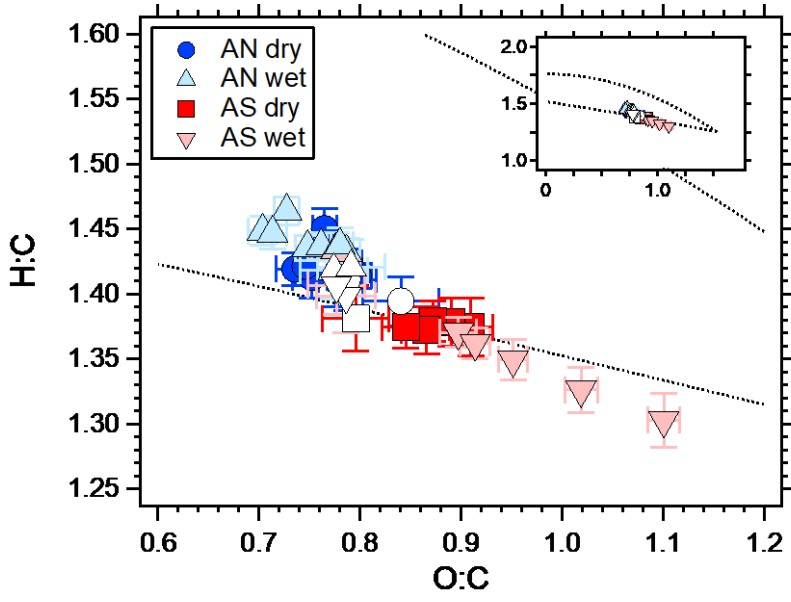





**Figure 6.** Van Krevelen diagram showing the SOA elemental ratios of the different experiments. The white markers represent SOA without seeds for corresponding symbol experiments. In general, O:C increased with increasing seed particle concentration. The dotted lines represent the Ng-triangle (Ng et al., 2011a) translated to the improved ambient elemental ratio parameterization (Canagaratna et al., 2015), to orientate the reader. The insert shows the same figure with different axis ranges, illustrating the fact that all data is within a relatively narrow range compared to e.g. ambient values.