# Peer review of "Effect of salt seed particle surface area, composition and phase on secondary organic aerosol mass yields in oxidation flow reactors"

_Atmospheric Chemistry and Physics, 2018_

## Referee Comment (RC1) · Anonymous Referee #1 · 11 Nov 2018

Ahlberg et al. presented a study that examined the effect of seed aerosol surface area, composition and phase on SOA mass yield in an oxidation flow reactor. This paper is potentially useful to the SOA community. However, there are portions of the manuscripts that are vague and confusing, and they need to be addressed before the manuscript can be considered for publication.

Specific comments:

1. Page 3 line 26: Why was a mixture of VOCs instead of pure a-pinene or pure m-xylene used in these experiments? Won't modeling condensational sinks be easier if single-component VOCs were used in the experiments? Different LVOCs will be lost

to the reactor's walls at different rates. Hence, the condensation sinks, extent of yield under-estimation etc. discussed in this manuscript are really some average of the a-pinene and m-xylene SOA systems. There is no way to determine which SOA system is the main cause of this yield under-estimation.

2. Page 3 line 29: Related to the previous comment, were the a-pinene and m-xylene present in equal quantities in the mixture used? The authors mentioned that the evaporation rate of the mixture decreased over time. Did the evaporation rates of a-pinene and m-xylene decrease at the same rates?

3. Page 4 line 11: I suggest that the authors define "OH reactivity" here (instead of a few lines further down) in order for their sentence to be less confusing.

4. Page 4 line 21: "from a few to 10 % on a mass basis..." it would be use to provide a number instead of using the term "a few".

5. Page 4 line 30: Please provide the reasoning behind drying the aerosols before measurements by the SMPS and AMS.

6. Page 6 line 11: How do the authors know that 100 % of the input VOCs will react in the flow reactor? VOC measurements? Modeling? It is worth explaining in the paper even though knowing the exact concentration of reacted VOC is likely not essential to the authors' comparison of relative increase in mass yields.

7. Page 6 line 17: The discussion on the mass yield uncertainties is confusing. What do the authors mean by "However, this may give a too small range"? Also, what do the authors mean by "No experiment was repeated fully"? I thought the authors replicated their experiments. Also, what do the authors mean by "..., but single point replicates gave a fractional error of 14 % compared to expected values"? What do these "expected values" refer to? In general, I highly recommend that the authors revise this discussion on mass yield uncertainties.

8. Page 6 line 24: The authors used the expression "error in the yields of these experiments". This use of this expression feels inappropriate since it gives readers an initial impression that errors were made in the calculation of the yields, which is not the intention of the authors. I suggest using the expression "extent of yield under-estimation" instead.

9. Page 6 line 27: What do the authors mean by "the produced organic sink"?

10. Page 6 line 28: How was the condensation sink calculated?

11. Page 6 line 31: The authors stated "This could be due to increased particle losses in the reactor. . .". Is this explanation valid given that the authors stated that particle losses are small in section 2?

12. Fig 4: The y-axis is given as "Yield/Max yield" while the main text states that the y-axis is Yield increase/Max yield increase. Which is the correct y axis label?

13. Page 7 line 16: The authors stated "At 10 ug/m3, the increase in yield is estimated from the linear regression between data points in Fig. 5. . .". This sentence is confusing. Which data points are the authors referring to? Do the authors fit all the data points for each VOC or only a subset of points near 10 ug/m3 OA? Why was linear regression used? Perhaps a non-logarithmic version of Fig. 5 should be provided because the current figure looks like it is better fitted to an exponential function.

14. Page 7 line 20: What do the authors mean by "a higher VOC produces a sink faster than a low input"? Are they referring to the condensation sink rate? Also, won't a higher VOC input result in higher concentrations of LVOC, some of which will be lost to the reactor's walls? Also, pervious environmental chamber studies have shown that the use of different VOC concentrations may result in different peroxy radical chemistry and hence different SOA mass yields. Did the authors consider this possibility in their flow reactor study?

15. Page 8 line 22: Do the authors mean "SOA mass fraction per particle"?

16. Page 8 line 25: Kroll and co-authors studied the heterogeneous OH photooxidation

of low-volatile organic aerosols such as squalane, oleic acid, linoleic acid etc., not SOA. Please make the appropriate revisions.

17. Have the authors looked at how the oxidation state changes as a function of SOA mass concentration? This analysis may provide some insights into the possibility that particle-phase chemistry may be the reason for the increase in O/C with SOA mass concentration.

18. Although this is likely the first study that examine the effect of seed aerosols on SOA formation in a flow reactor, the effect of seed aerosols on SOA formation in chamber experiments has been studied by Huang et al. (JGR: Atmospheres, 2016). In that study, the authors performed PMF analysis on their data to determine that differences in particle-phase chemistry contributed to the differences in SOA mass yields when different seed aerosols were used in a-pinene ozonolysis experiments. Perhaps the authors can perform similar PMF analysis to their data to explain their results in this flow reactor study.

---

## Referee Comment (RC2) · Anonymous Referee #2 · 14 Dec 2018

Ahlberg et al. investigated the yield of secondary organic aerosol (SOA) generated from the photooxidation of an m-xylene/$\alpha$-pinene mixture in an oxidation flow reactor (OFR) as a function of seed particle surface area, type, and phase. At the operating conditions that were used, they found that the SOA yield increased by a factor of 2-3.5 as a function of increasing seed concentration from 0 to 3000 um$^2$ cm$^{-3}$ (condensation sink = 0 to 0.1 s$^{-1}$), with the highest increases observed in the presence of ammonium nitrate and wet ammonium sulfate seeds. The seed dependence on SOA yield was modeled using the "LVOC fate correction" model published by Palm et al., ACP, 2016, and hypothesized to be due to residence-time-limited condensation of oxidation products in the OFR. The effect of SOA elemental ratios (H:C and O:C) was also characterized as a function of seed particle concentration. Given the emergence of OFRs as a technique to characterize SOA formation, and the previously established dependence of SOA yields on available condensation sink, I would support publication of this manuscript in ACP after consideration of my comments below.

1. Somewhere in the manuscript, perhaps in conclusions, the authors should add a paragraph to discuss recommendations for (i) seed particle composition and (ii) seed particle surface area concentration for SOA yield measurements in OFR studies. For example:
   a. Should ammonium nitrate seeds be used because ammonium nitrate is less humidity-dependent than ammonium sulfate? Are there deficiencies with using either AN or AS that could be addressed with other seed types?
   b. It seems that CS > 0.06 – 0.07 s$^{-1}$ is required at the conditions that were used in these studies (e.g. PAM, 160 sec residence time). It would be useful to model or parameterize the CS that is required, as a function of residence time, in graphical or equation form.
   c. The work would have greater impact if the result can be generalized to other OFR designs that might for example have different surface-to-volume ratio than the PAM reactor and consequently different CS required to promote condensation.

2. I think it would be better to plot Figure 2 later in the paper so that the fractional LVOC loss pathways for the data points shown in Fig. 3 can be added to Figure 2. Additionally, the justification for assuming an OH reaction rate of 1E-11 cm$^3$/molec/sec is never provided; thus, the associated level of uncertainty in the modeling is unclear. At the very least the model should be run over a wider range of OH rate constants—perhaps a factor of 2 or 5 in either direction relative to 1E-11 –to evaluate the sensitivity of the model result to the specified kOH.

3. I prefer that the authors not refer to their operating conditions as "atmospherically relevant." It's already implied that atmospherically relevant conditions were used, but the reader can decide the atmospheric relevance themselves in the context of specific atmospheric conditions or source regions.  In the instances below, for example, it may not be generally true that RH = 60%, ORG = 5 ug/m3 are always "atmospherically relevant" conditions:
   a. "using an oxidation flow reactor operated at an **atmospherically relevant RH of 60** %" (P1, L12-L13)
   b. "A relatively low, and **atmospherically relevant, SOA mass concentration of ~5 μg m$^{-3}$**" (P3, L27)

c. "liquid-liquid phase separation rarely occur at O:C ratios higher than 0.7 in systems containing **atmospherically relevant** organics" (P8, L9-L10)

4. Introduction, and elsewhere as appropriate: Jathar et al., ES&T, 2017  and Zhao et al., ES&T, 2018 should be cited and discussed as their work also investigated SOA yield dependence as a function of particle concentration / condensation sink.

5. P1, L18: "low volatile" → "low volatility"

6. P2, L24: I suggest adding "the yields of **SOA generated from** m-xylene…" or similar.

7. P3, L28: Please show a representative subset of SMPS & AMS size distributions of the seed/SOA mixtures as a function of seed concentration. How much of the SOA condenses on the seeds and how much homogenously nucleates to form new particles? This is not clear at present. The size distributions could be shown in the supplement.

8. P5, L6: The authors state: "Before experiments, the reactor was run without seeds or VOCs until the volume concentration was below 0.2 $\mu m^3$ cm-3". Please clarify whether the lights were turned on for generating $OH/O_3$ during these periods?

9. P6, L5: What is the mass-/volume-weighted mobility diameter? (possibly the more critical value in assessing how much of the mass is not transmitted through the AMS inlet lens?)

10. P7, L1-2:  This statement is not clear: "which is similar the residence time of the reactor shortcircuit"

11. P8, L9: problem with citation formatting.

12. P8, L20-L24: The authors state: "*The O:C value increases with increasing SOA mass concentration (and seed particle concentration since these are connected), which is opposite to what is expected since more oxidized molecules tend to be less volatile (Shilling et al. 2009).[…] Increasing the seed particle number concentration also decreases the SOA mass fraction. It follows that SOA mass then is spread out on more particles, leaving less organics per particle, which could enhance the partitioning of more volatile material to the gas phase. This would leave more time for gas phase oxidation, and consequently a higher O:C ratio, provided the molecules partition to the particles.*" There are a lot of nuances to this discussion. At the least, it is unclear whether the logic that is presented here is fully supported by the measurements. It also seems self-contradictory-for example, how can higher seed particle concentrations promote condensation of oxidized organic vapors (the main result of the paper) while simultaneously "[enhancing] the portioning of more volatile material to the gas phase"? These statements need to be reevaluated and modified in order to provide a more cohesive interpretation of the data.

13. Figure 3-4 and 6 should indicate the SOA precursors that were used (aPinene + m-Xylene), preferably in the figure axis labels or captions or both.

14. Figure 4: what are the propagated uncertainties in yield ratios, condensation sink and in LVOC fate model outputs?

**References**

S. H. Jathar, B. Friedman, A. A. Galang, M. F. Link, P. Brophy, J. Volckens, S. Eluri, and D. K. Farmer. Linking Load, Fuel and Emission Controls to Photochemical Production of Secondary Organic Aerosol from a Diesel Engine. *Environ Sci. Technol*., 51(3), 1377-1386, 2017.

Y. Zhao, A. T. Lambe, R. Saleh, G. Saliba, and A. L. Robinson. Secondary Organic Aerosol Production from Gasoline Vehicle Exhaust: Effects of Engine Technology, Cold Start, and Emissions Certification Standard. *Environ. Sci. Technol.*, *52* (3), 1253–1261, DOI**:** 10.1021/acs.est.7b05045, 2018.

---

## Author Comment (AC1) · 26 Jan 2019

We sincerely thank both reviewers for valuable and relevant feedback that challenged our interpretations and helped improve the manuscript. Below are the copied comments of the reviewer reports (Calibri, highlighted in yellow) and our corresponding answers (in black) and changes in the manuscript *(italic)*.

**Anonymous Referee #1**

Ahlberg et al. presented a study that examined the effect of seed aerosol surface area, composition and phase on SOA mass yield in an oxidation flow reactor. This paper is potentially useful to the SOA community. However, there are portions of the manuscripts that are vague and confusing, and they need to be addressed before the manuscript can be considered for publication.

Thank you for the comments and effort. With the answers below, we hope that we have clarified any vagueness and confusing parts.

Specific comments:
1. Page 3 line 26: Why was a mixture of VOCs instead of pure a-pinene or pure mxylene used in these experiments? Won't modeling condensational sinks be easier if single-component VOCs were used in the experiments? Different LVOCs will be lost to the reactor's walls at different rates. Hence, the condensation sinks, extent of yield under-estimation etc. discussed in this manuscript are really some average of the apinene and m-xylene SOA systems. There is no way to determine which SOA system is the main cause of this yield under-estimation.

Using a mixture was not important for the end results. The reason we choose this mixture instead of just one VOC is that it may resemble ambient conditions more (where both anthropogenic and biogenic SOA are present in a mixture). Further, we showed in a previous paper (Ahlberg et al., 2017) that the SOA formation from this VOC mixture can be represented with a model that assumes that the condensed phase is an ideal mixture (following Raoult's law). If we would have seen significant deviations from what can be expected from an ideal mixture in our previous work, perhaps our approach would have been different in the present study. Yes, the yield under-estimation is an average of the two SOA systems. But as it is a function of the condensation sink, this doesn't matter for our conclusions regarding the yields. The condensation sink is a measured quantity in most OFR experiments. Although two VOCs introduces more complexity, both VOCs chosen are well known SOA precursors. Also, the wall loss effect of single SOA precursor experiments is very complex to model in detail. Even from one single VOC the oxidation products consists a myriad of different molecules with a wide range of different pure liquid saturation vapour pressures, reactivity in the condensed phase and on the walls, and molecular diffusion coefficients. All these properties influence their wall losses in OFRs.

2. Page 3 line 29: Related to the previous comment, were the a-pinene and m-xylene present in equal quantities in the mixture used? The authors mentioned that the evaporation rate of the mixture decreased over time. Did the evaporation rates of a-pinene and m-xylene decrease at the same rates?

The VOC concentrations were pretty similar (5.2 and 6.7 ppb respectively). This is already stated in the last paragraph of section 2.1. The evaporation rates declined by 10 and 17 % during the second week of the weighing experiment. This was 14 days after the start of the weighing experiment. As stated in the manuscript, it is important to note, that the timescales during weighing are much longer than the time-scales of the SOA experiments.

3. Page 4 line 11: I suggest that the authors define "OH reactivity" here (instead of a few lines further down) in order for their sentence to be less confusing.

Agree. Moved the definition.

4. Page 4 line 21: "from a few to 10 % on a mass basis. . ." it would be use to provide a number instead of using the term "a few".

Since we didn't measure the losses it is hard to give numbers. But our main point is that we don't care about particle losses at the inlet, and that we therefore can assume losses lower than 10%.

We changed the formulation to:

*Particle losses depend on reactor settings and particle sizes, but are generally lower than 10 % on a mass basis (Martinsson et al., 2015; Karjalainen et al., 2016; Ortega et al., 2016; Palm et al., 2016).*

**5. Page 4 line 30: Please provide the reasoning behind drying the aerosols before measurements by the SMPS and AMS.**

Drying of the aerosols before the measurements by the SMPS is generally seen as crucial since one otherwise measures wet particle sizes, which varies with RH. For the AMS, which measures chemically resolved particle mass, drying is recommended to reduce uncertainties in the collection efficiency. The particles may bounce less if they are wet and also, the inlet losses are a function of size which may depend on the RH.

**6. Page 6 line 11: How do the authors know that 100 % of the input VOCs will react in the flow reactor? VOC measurements? Modeling? It is worth explaining in the paper even though knowing the exact concentration of reacted VOC is likely not essential to the authors' comparison of relative increase in mass yields.**

This was calculated from the calibrated OH exposure and tabulated reaction rates of the VOCs. OH exposure divided by time gives OH concentration if it is assumed that OH is stable throughout the reactor volume. This is a simplification, but Li et al. (2015) showed that this is not too far from the truth. Since the OH loss rate is also low at the inlet, OH concentration builds up very quickly. With our OH exposure (7e11 molecs cm$^{-3}$ s), the concentration was 4.3e9 molecules cm$^{-3}$. This gives lifetimes for α-pinene and m-xylene of 4.3 and 16.2 seconds respectively. Compared to the residence time (160 s), this is very short. We clarified the manuscript as follows.

Old text: In the oxidizing environment of the reactor, 100 % of the input VOCs are expected to react.

New formulation: *Assuming constant OH concentration, with our settings the VOC lifetimes for reaction with OH are short compared to the residence time (4.3 and 16.2 seconds, for α-pinene and m-xylene respectively). Therefore we assume that all VOCs have reacted.*

**7. Page 6 line 17: The discussion on the mass yield uncertainties is confusing. What do the authors mean by "However, this may give a too small range"? Also, what do the authors mean by "No experiment was repeated fully"? I thought the authors replicated their experiments. Also, what do the authors mean by ". . ., but single point replicates gave a fractional error of 14 % compared to expected values"? What do these "expected values" refer to? In general, I highly recommend that the authors revise this discussion on mass yield uncertainties.**

We agree that this was a bit confusing. We repeated the SOA only experiment twice per day. This is the most important test, since it is very sensitive to changes in OH exposure, VOC evaporation rate and flows. Also, this determines the VOC concentration during the full experiment and consequently the yields with seed particles. We have rewritten the paragraph to make it more clear.

Old text: The uncertainty in the yield increase was calculated from the standard deviation of the measurements. The fractional uncertainty was between 6-10 %. However, this may give a too small range. Although all flow, pressure, and OFR settings were checked, a variation larger than the experiment standard deviation is expected since the setup was highly sensitive to small perturbations. No experiment was repeated fully, but single point replicates gave a fractional error of 14 % compared to expected values. Therefore, a conservative expected repeatability of the experiments is within 20 %.

New text: *The uncertainty in the yield increase was calculated from error propagation of the standard deviations of the measurements. The fractional uncertainty with this method was between 6-10 %. However, this only reflects the precision in one experiment. Although all flow, pressure, and OFR settings were checked repeatedly, a variation larger than single experiment standard deviation is expected since the setup was highly sensitive to small perturbations. No experiment was repeated fully, but SOA levels without seeds was tested twice per day. Replicates of SOA yield with seeds gave a fractional error of 14 % compared to previous values. Therefore, a conservative expected repeatability of the experiments is within 20 %.*

**8. Page 6 line 24: The authors used the expression "error in the yields of these experiments". This use of this expression feels inappropriate since it gives readers an initial impression that errors were made in the calculation of the yields, which is not the intention of the authors. I suggest using the expression "extent of yield under-estimation" instead.**

Fixed. The sentence now reads:

*The extent of yield underestimation in these experiments can be calculated by normalizing the yield increase with the maximum yield increase.*

**9. Page 6 line 27: What do the authors mean by "the produced organic sink"?**

It is the contribution of SOA to the CS. We modified the sentence to:

*Also seen in Fig. 4 is the modeled bias (fraction condensed on particles), following a similar trend but slightly lower since the experimental data only considers the condensation sink of the seed particles, while in the model the total sink (seed + SOA) is taken into account.*

10. Page 6 line 28: How was the condensation sink calculated?

The condensation sink was calculated as in Pirjola et al. (1999). For this calculation, we made assumptions of the diffusion and mass accommodation coefficients following Palm et al. (2016). The diffusion coefficient was 7e-6 $m^2$ $s^{-1}$ based on an oxidized 200 g $mol^{-1}$ (Tang et al., 2015) molecule and the accommodation coefficient was set to 1 (Julin et al., 2014). See also answers to referee #2. The following was added to the manuscript:

Section 2.2: *For model sensitivity tests and uncertainties, the reader is referred to the original paper.*

11. Page 6 line 31: The authors stated "This could be due to increased particle losses in the reactor. . .". Is this explanation valid given that the authors stated that particle losses are small in section 2?

You are right that this looks contradictory. And even though losses of 10% could affect the curves, the losses shouldn't change much since the particle sizes will remain pretty much the same. Instead it is more likely that it could be due to the SOA being spread out on more particles which could increase heterogeneous oxidation that leads to fragmentation. But we don't want to speculate too much on this difference since it is not so pronounced. The manuscript was changed as follows:

Old text: However, while the LVOC model yield continues to increase with increased seed area concentration, the increase in the experimental yields levels off. This could be due to increased particle losses inside the reactor, which have not been considered.

New text: However, while the LVOC model yield continues to increase with increased seed area concentration, the increase in the experimental yields levels off. *This could be due to increased fragmentation losses from heterogeneous oxidation, since a larger portion of the SOA will be exposed with higher seed particle concentration.*

12. Fig 4: The y-axis is given as "Yield/Max yield" while the main text states that the y-axis is Yield increase/Max yield increase. Which is the correct y axis label?

This is actually the same thing (yield increase is the ratio of yield to SOA only yield and taking two yield increase ratios the SOA only yield is deleted). To keep it simple and avoid confusion we deleted "increase" in the text.

New text: *The extent of yield underestimation in these experiments can be calculated by normalizing the yield with the maximum yield.*

13. Page 7 line 16: The authors stated "At 10 ug/m3, the increase in yield is estimated from the linear regression between data points in Fig. 5. . .". This sentence is confusing. Which data points are the authors referring to? Do the authors fit all the data points for each VOC or only a subset of points near 10 ug/m3 OA? Why was linear regression used? Perhaps a non-logarithmic version of Fig. 5 should be provided because the current figure looks like it is better fitted to an exponential function.

Since we don't have data for exactly 10 ug/m3, we estimate it by simply taking two adjacent points and drawing a line between them. Yes, the full range of the data is better with fitted with other methods, but doing this would be unnecessary for this exercise which is a rough estimate

We changed the phrasing to:

*At 10 μg m-3 the increase in yield is estimated, from linear regression between adjacent datapoints in Fig. 5, at 67 %, 80 %, 24 % and 94 % for α-pinene, m-xylene, myrcene and isoprene respectively*

14. Page 7 line 20: What do the authors mean by "a higher VOC produces a sink faster than a low input"? Are they referring to the condensation sink rate? Also, won't a higher VOC input result in higher concentrations of LVOC, some of which will be lost to the reactor's walls? Also, pervious environmental chamber studies have shown that the use of different VOC concentrations may result in different peroxy radical chemistry and hence different SOA mass yields. Did the authors consider this possibility in their flow reactor study?

We are referring to the condensation sink, yes. Given that the OH concentration is not affected, a high concentration of VOCs would produce a certain amount of SOA faster than a low concentration. No, we did not consider the peroxy radical chemistry. A recent paper (Peng et al., 2019) discusses this issue in OFRs, but we

don't it is necessary to include this in the present manuscript since the VOC concentration was kept constant in all experiments.

The text was clarified as follows:

*...since higher VOC concentrations also produce a condensation sink faster than a low input VOC concentration*

15. Page 8 line 22: Do the authors mean "SOA mass fraction per particle"?

Yes. Modified the text as per suggestion.

16. Page 8 line 25: Kroll and co-authors studied the heterogeneous OH photooxidation of low-volatile organic aerosols such as squalane, oleic acid, linoleic acid etc., not SOA. Please make the appropriate revisions.

Thanks for noting this mistake. We changed "SOA" to "organic particles."

17. Have the authors looked at how the oxidation state changes as a function of SOA mass concentration? This analysis may provide some insights into the possibility that particle-phase chemistry may be the reason for the increase in O/C with SOA mass concentration.

Yes, but since H:C is within such a small range, and OSc=2*O:C-H:C, it doesn't add information that is not already in the van Krevelen diagram.

18. Although this is likely the first study that examine the effect of seed aerosols on SOA formation in a flow reactor, the effect of seed aerosols on SOA formation in chamber experiments has been studied by Huang et al. (JGR: Atmospheres, 2016). In that study, the authors performed PMF analysis on their data to determine that differences in particle-phase chemistry contributed to the differences in SOA mass yields when different seed aerosols were used in a-pinene ozonolysis experiments. Perhaps the authors can perform similar PMF analysis to their data to explain their results in this flow reactor study.

The suggestion to perform positive matrix factorization (PMF) on the recorded mass spectra is intriguing. However, it falls beyond the scope of the present study. The operational principle of PMF is that is exploits co-variance between measured variables between samples to construct static "building blocks" of the measured signal (factors), with varying abundance between samples. As applied to aerosol mass spectrometry, this means one reconstructs the measured ion time series by different combinations of factor mass spectra. This is readily done on dynamic data such as ambient measurement, or as for Huang et al, batch mode oxidation experiments (where an aerosol is continually measured in real time as it undergoes physicochemical transformation). OFR experiments of the type reported here are fundamentally different as the temporal variability is under the detailed control of the experimentalist; one changes set points as steady state and sufficient counting statistics are obtained. Therefore, PMF cannot be used in the conventional manner for our dataset.

**Anonymous Referee #2**

Ahlberg et al. investigated the yield of secondary organic aerosol (SOA) generated from the photooxidation of an m-xylene/⍺-pinene mixture in an oxidation flow reactor (OFR) as a function of seed particle surface area, type, and phase. At the operating conditions that were used, they found that the SOA yield increased by a factor of 2-3.5 as a function of increasing seed concentration from 0 to 3000 um2 cm-3 (condensation sink = 0 to 0.1 s-1), with the highest increases observed in the presence of ammonium nitrate and wet ammonium sulfate seeds. The seed dependence on SOA yield was modeled using the "LVOC fate correction" model published by Palm et al., ACP, 2016, and hypothesized to be due to residence-time-limited condensation of oxidation products in the OFR. The effect of SOA elemental ratios (H:C and O:C) was also characterized as a function of seed particle concentration. Given the emergence of OFRs as a technique to characterize SOA formation, and the previously established dependence of SOA yields on available condensation sink, I would support publication of this manuscript in ACP after consideration of my comments below.

Thank you for the comments and effort.

1. Somewhere in the manuscript, perhaps in conclusions, the authors should add a paragraph to discuss recommendations for (i) seed particle composition and (ii) seed particle surface area concentration for SOA yield measurements in OFR studies. For example:
a. Should ammonium nitrate seeds be used because ammonium nitrate is less humiditydependent than ammonium sulfate? Are there deficiencies with using either AN or AS that could be addressed with other seed types?

We don't believe AN is less humidity-dependent than AS in terms of growth factor. However, efflorescence is not observed in AN which will affect the results depending on RH and seed particle drying. The second paragraph of the conclusions addresses this point, and includes recommendations for future studies on this issue.

b. It seems that CS > 0.06 – 0.07 s-1 is required at the conditions that were used in these studies (e.g. PAM, 160 sec residence time). It would be useful to model or parameterize the CS that is required, as a function of residence time, in graphical or equation form.
c. The work would have greater impact if the result can be generalized to other OFR designs that might for example have different surface-to-volume ratio than the PAM reactor and consequently different CS required to promote condensation.

Both of these are very good points that we tried to implement in a simplified manner. However, it is not easy to represent all different conditions and interdependent variables in neither graphical or equation form. E.g. the coefficient of eddy diffusion, $k_e$, used to calculate the wall loss, is dependent on reactor volume. But several different reactor designs could have the same volume but with different surface to volume (A/V) ratios, so a graphical representation of CS needed as a function of surface to volume and residence time would need assumptions of the relationship between radius and length, which is precisely the variables that are altered for different designs. Further, a representation of CS needed as a function of residence time would need to assume constant OH exposure or constant UV lamp power (actinic flux). Both of these are usually changed between different experiments (OH exposure follows the residence time if lamp power is constant). So, a generalization could be made if we make assumptions of certain variables, but this could be seen as an oversimplification that would not benefit the OFR community. It could introduce more confusion and we strongly encourage each group to model their specific settings for each experiment. The following changes in the manuscript have been made:

Old text: The yield increase leveled off at a dry seed particle condensation sink of ~0.05 s$^{-1}$ corresponding to a surface area concentration of ~1600 µm$^2$ cm$^{-3}$. This implies that it is crucial that the condensation sink is accounted for in OFR experiments where the absolute SOA mass is of interest.

Now reads: *The yield increase leveled off at a dry seed particle condensation sink of ~0.05 s$^{-1}$ corresponding to a surface area concentration of ~1600 µm$^2$ cm$^{-3}$. This value will be different for different reactor geometries and settings (such as OH exposure and residence time) and implies that it is crucial that the condensation sink is evaluated in all OFR experiments where the absolute SOA mass is of interest.*

2. I think it would be better to plot Figure 2 later in the paper so that the fractional LVOC loss pathways for the data points shown in Fig. 3 can be added to Figure 2.

If we understand correctly, this is what is shown in figure 4 where we combined model and data.

Additionally, the justification for assuming an OH reaction rate of 1E-11 cm3/molec/sec is never provided; thus, the associated level of uncertainty in the modeling is unclear. At the very least the model should be run over a wider range of OH rate constants—perhaps a factor of 2 or 5 in either direction relative to 1E-11 –to evaluate the sensitivity of the model result to the specified kOH.

The rate constant was taken from the original model in (Palm et al., 2016), where they assumed an oxygenated molecule with 10 carbon atoms and no carbon double bonds and took values from (Ziemann and Atkinson, 2012). In the alternative version of figure 4 shown below we added modelled SOA yield for rate constants of 2e-11 and 0.5e-11 cm$^3$ molecs$^{-1}$ s$^{-1}$.

[Figure]

As can be seen, the graphical agreement between measurements and model can be improved by using a slower rate constant. This could however be for the wrong reason. First of all, the datapoints are normalized to the maximum yield so they reach the value of one, which is not the case for the model which reaches ~0.9 condensed fraction at the maximum CS used. Also, the model takes into account any CS, while the data only used the seed CS. The purpose of the original figure was to show the similarities of the model and all measurements when they are normalized.

Using just one rate constant is of course a simplification, but from the sensitivity analysis that was done by Palm et al. (2016) the conclusion was that the model is more sensitive to other parameters, namely the sticking coefficient, the condensation sink and the average residence time. Therefore, we don't wish to publish the above figure, and instead the reader should refer to the original model. We modified the caption of figure 2 as follows:

Old text: Figure 2. The fractional fate of LVOCs as a function of particle surface area concentration, using the model of Palm *et al.* (2016). A molecule is assumed to fragment after reacting with OH five times. OH reaction rate was $1 \times 10^{-11}$ $cm^3$ $molecs^{-1}$ $s^{-1}$. For reactor settings, see text.

Now reads: *Figure 2. The fractional fate of LVOCs as a function of particle surface area concentration, using the model of Palm et al. (2016), with the same OH reaction rate ($1 \times 10^{-11}$ $cm^3$ $molecs^{-1}$ $s^{-1}$) and assuming fragmentation after reaction with OH five times. For reactor settings, see text.*

3. I prefer that the authors not refer to their operating conditions as "atmospherically relevant." It's already implied that atmospherically relevant conditions were used, but the reader can decide the atmospheric relevance themselves in the context of specific atmospheric conditions or source regions. In the instances below, for example, it may not be generally true that RH = 60%, ORG = 5 ug/m3 are always "atmospherically relevant" conditions:

We agree.

a. "using an oxidation flow reactor operated at an atmospherically relevant RH of 60 %" (P1, L12-L13)

Deleted "atmospherically relevant."

b. "A relatively low, and atmospherically relevant, SOA mass concentration of ~5 µg m-3" (P3, L27)

Deleted "atmospherically relevant."

c. "liquid-liquid phase separation rarely occur at O:C ratios higher than 0.7 in systems containing atmospherically relevant organics" (P8, L9-L10)

Deleted "atmospherically relevant."

4. Introduction, and elsewhere as appropriate: Jathar et al., ES&T, 2017 and Zhao et al., ES&T, 2018 should be cited and discussed as their work also investigated SOA yield dependence as a function of particle concentration / condensation sink.

We have added these references at the following places:

Introduction: *Due to the fast processing in flow reactors, several studies have discussed the potential problem with low condensation sinks resulting in lower yields (Lambe et al., 2015; Palm et al., 2016; Ahlberg et al., 2017; Jathar et al., 2017; Simonen et al., 2017; Zhao et al., 2018).*

Section 3*: In all experiments the yield increased significantly with seed particle surface area, confirming previous findings (Lambe et al., 2015; Palm et al., 2016; Ahlberg et al., 2017; Jathar et al., 2017; Zhao et al., 2018).*

5. P1, L18: "low volatile" → "low volatility"

Changed.

6. P2, L24: I suggest adding "the yields of SOA generated from m-xylene…" or similar.

Old text: In similar studies the yields of m-xylene…

Now reads: *In similar studies the SOA yields of m-xylene…*

7. P3, L28: Please show a representative subset of SMPS & AMS size distributions of the seed/SOA mixtures as a function of seed concentration. How much of the SOA condenses on the seeds and how much homogenously nucleates to form new particles? This is not clear at present. The size distributions could be shown in the supplement.

The nucleation mode clearly disappeared after adding a certain amount of seed particles. We agree that we should show this in the manuscript. We did not make use of the AMS pToF data. The following figure and caption was added to the supplement:

[Figure]

*Figure S2. Example of how the volume size distribution changed during and experiment, from purely nucleated particles consisting of only SOA, through a mixture of nucleated and seed particles, to purely seed particles with condensed organics. Y axis is normalized so the differences in Dp are more clear.*

The manuscript was modified as follows (section 2.4):

*An example of the volume size distributions during an experiment is shown in figure S2.*

8. P5, L6: The authors state: "Before experiments, the reactor was run without seeds or VOCs until the volume concentration was below 0.2 µm3 cm-3". Please clarify whether the lights were turned on for generating OH/O3 during these periods?

The sentence now reads:

Before experiments, the reactor was run *with the lamps on* without seeds or VOCs until the volume concentration was below 0.2 µm3 cm-3, as measured by the SMPS.

9. P6, L5: What is the mass-/volume-weighted mobility diameter? (possibly the more critical value in assessing how much of the mass is not transmitted through the AMS inlet lens?)

This is answered by the figure we added in comment nr 7 (fig. S2). The manuscript (section 2.4) was modified as follows (italic):

The relatively low CE of SOA may not only be a bounce effect, since these particles were significantly smaller, with a number mode around 20 nm, *and a volume mode around 40-50 nm in mobility diameter*, which to a higher degree are lost in the aerodynamic lens inlet of the AMS.

10. P7, L1-2: This statement is not clear: "which is similar the residence time of the reactor shortcircuit"

In some reactors, a short circuit bypasses the rest of the reactor. Material travelling through this way will have a significantly shorter residence time. This has been seen in for the PAM OFR in the references stated in the manuscript. The flow pattern in the PAM OFR was recently explored in more detail in Mitroo et al. (2018).

11. P8, L9: problem with citation formatting.

Actually, we don't see this…

12. P8, L20-L24: The authors state: "The O:C value increases with increasing SOA mass concentration (and seed particle concentration since these are connected), which is opposite to what is expected since more oxidized molecules tend to be less volatile (Shilling et al. 2009).[…] Increasing the seed particle number concentration also decreases the SOA mass fraction. It follows that SOA mass then is spread out on more particles, leaving less organics per particle, which could enhance the partitioning of more volatile material to the gas phase. This would leave more time for gas phase oxidation, and consequently a higher O:C ratio, provided the molecules partition to the particles." There are a lot of nuances to this discussion. At the least, it is unclear whether the logic that is presented here is fully supported by the measurements. It also seems self-contradictory-for example, how can higher seed particle concentrations promote condensation of oxidized organic vapors (the main result of the paper) while simultaneously "[enhancing] the portioning of more volatile material to the gas phase"? These statements need to be reevaluated and modified in order to provide a more cohesive interpretation of the data.

We agree that this discussion is a bit speculative and unclear. Although one could imagine the same SOA yield despite more oxidation, it is contradictory to assume a larger evaporation rate and larger SOA yield simultaneously. It is more likely that it is the effect discussed in answers to referee #1 comment 11: A larger SOA area is exposed to heterogeneous oxidation with higher seed particle number concentrations.

We changed the paragraph as follows.

[revised manuscript text omitted]